# The Role of Critical Thinking in Predicting and Improving Academic Performance

**Silvia F. Rivas** [1,*] **, Carlos Saiz** [1] **and Leandro S. Almeida** [2]

1   Psychology Faculty, University of Salamanca, 37005 Salamanca, Spain
2   Institute of Education, University of Minho, 4710-057 Braga, Portugal
*   Correspondence: silviaferivas@usal.es

**Abstract:** Today's world demands new ways of dealing with problems and different ways of preparing for them. Some studies argue that these new demands also require new skills. Critical thinking (CT) involves a set of skills that are entirely relevant to today's adaptive needs. In this study, we explore the extent to which CT serves to both account for and improve academic performance. To do this, we measured the CT skills of a number of undergraduate students, along with their university admission grades and average course grades. We successfully established the structural validity of the general construct of CT, along with a strong relationship between CT measurements at two different timepoints (at the beginning and end of studying a critical thinking subject area) and the admission grade and average grade variables. These data lead us to reflect on using CT level as a suitable assessment of academic performance. We also consider the limits of our findings and their implications.

**Keywords:** higher education; critical thinking; academic performance; assessment

## 1. Introduction

The world has probably seen more changes in recent decades than at any time since the industrial revolution. These changes demand substantial transformation in society in general, and particularly in our educational institutions. This transformation is characterised, above all, by increasingly complex problems, caused, in large part, by changes brought about by information and communication technologies (ICT) and economic competitiveness in a globalised world. All this means that today's issues are a mix of the social, the professional and the personal. There used to be a clear line separating these types of issues and situations, but today, it is very much blurred.

These changes mean that people today must increasingly develop their abilities to make sound decisions and solve problems effectively. At the same time, today's complex real-world problems are not solved by following a model of intelligence based primarily on classic IQ or logical reasoning [1–4]. Socio-emotional dimensions of intelligence, along with competences in creative and critical thinking, are fundamental today in the cognitive exercises of problem-solving and problem-finding [5–9].

Tackling the problems of today's world requires us to employ more critical thinking (CT) skills, as they seem to be specifically designed for these problems, given that they involve skills and strategies that are more generalisable or transferable than others [10]. CT skills being mainly horizontal and contextualised competences means that we can use them equally effectively in many situations. The flexibility and adaptability of CT to different situations and contexts makes it an excellent candidate for handling today's changes and new demands. In this sense, education and training at all levels need to adapt much more quickly to these changes. In particular, university education is called upon to change further in order to train young people in their transition to adulthood [11]. Universities need to transform themselves more than other institutions in society if they are to progress, lead the necessary change and produce professionally trained, mature, socially responsible adults.

However, there seems to be a worrying disconnect between academia and the real world. Companies are increasingly demanding more transversal or horizontal competences. For example, they need not only qualified biologists or engineers, but also professionals who, from their specialist area (vertical competences), solve problems in different work contexts, make decisions individually and collectively and communicate their results in an argumentatively precise manner [12]. These general skills in solving decontextualised problems, making good decisions and arguing persuasively are not the focus of today's higher education. As we noted above, developing these competences requires a different conception of our cognitive machinery and the skills that are really fundamental. Additionally, and no less important, it requires an awareness of the deficiencies and limitations of that machinery.

Higher education today suffers from three significant ills [12]. As implied earlier, the first of these stems from the fact that the real world has changed faster than universities have. The second relates to the fact that the predominant model is still the administration of accumulated knowledge and not the management of learning, in terms of the process of acquiring knowledge and developing competences. The third is that training is still mainly vertical or disciplinary and rarely horizontal or cross-disciplinary. We have already noted that in the real world—the world of work—companies and some institutions require qualified people who can perform in very different contexts and solve problems equally well in any of them. Our students lack sufficient preparation for working with real problems in different contexts—in other words, the widespread application of knowledge and its inter-domain practice is rather limited, at least in comparison to what it needs to be. With these three dominant characteristics, graduates are unlikely to be qualified to perform and solve problems in complex, new, changing contexts.

New times and new problems require new tools and new strategies. Today's world has become so complicated that the need for lifelong learning is now a given. One real and challenging change is therefore the ongoing preparation of professionals and citizens. The consequence of this is the issue of institutions offering courses throughout people's lives. This brings us to a reflection that has yet to be properly addressed: being aware of the differences between teaching, learning and training helps us to better address this issue. If it is not possible to offer education for all ages, yet lifelong learning is needed, how can this be achieved? In our view, the answer is primarily training, and not just apprenticeships. Training implies student autonomy and almost complete independence. This means that they know where to go for documentation; know how to apply reliable criteria when evaluating what they find; and are able to reach their own conclusions and explanations about the problems or questions that interest them. This reflection therefore relates to the question of where training can or should be given today.

However, it is important to note that some strategies are particularly effective in lifelong or ongoing training. People must learn how to learn, and learn to develop behavioural, cognitive and motivational strategies that regulate their ongoing acquisition of knowledge and skills. At the same time, with a greater emphasis on training than on learning, people need to be involved in processes that encourage their autonomy, initiative and responsibility at personal, professional and social levels. Three UNESCO reports, spaced roughly two decades apart [13–15], show attempts to adapt to the changes that have taken place at each point in time.

The latest UNESCO report "retires the Delors report and redraws a new horizon for education" [16]. According to Sobhi Tawil, a leading member of the commission that drafted the report: "In the current global debate on the future of education, there are two parallel currents. The first concerns the role of education in the post-2015 international development agenda. The second relates to the way in which the transformation of society globally impacts on our approach to education and learning..." [17]. The second current is that which today requires training in terms of learning to understand, to do, to be, and to be with others.

If changes are made in educational methodologies by giving greater prominence to learning processes and focusing on greater application and cross-domain practice, then higher education can be made to take on the development of critical thinking as its main objective [12]. Indeed, today's changes and demands are greater than in past decades, and how we deal with them needs to be through competences and strategies that can be employed to handle this complexity.

CT offers what is needed in terms of how to progress, train and generalise. Being able to change faster entails developing collaborative communities of enquiry that participate in solving real projects together. In training, it is important to consider different methods that facilitate the acquisition processes, such as working on tasks involving not only comprehension but also production, aimed above all at acquiring knowledge based on explanation, and applying this knowledge in different contexts. In order to be able to generalise in different situations or contexts, it is essential to develop CT's fundamental competences—such as explanation (search for causality), decision-making and problem-solving. These skills are domain-independent, meaning they are required skills that need to be used throughout any situation or context [12].

Before specifying the research problem and the hypotheses, it is worth remembering the conceptual approach published in different works, some of which have been cited above. We understand that "to think critically is to reach the best explanation for a fact, phenomenon or problem, in order to gain insight and solve it effectively" ([18], p. 27). Knowing or solving a problem requires knowing what causes are responsible for certain events or problems. If the explanation is sound, we can choose the best course of action or the best option to resolve the situation effectively by bringing about the desired change. Therefore, it is the explanation that determines the decision and the solution and, finally, the change and well-being or achievement (a full description of our approach can be found at [19]). If improving critical thinking is our goal, we believe that this is the best way to achieve it. Therefore, our intervention aims to achieve change through the development of the aforementioned skills. This is the intervention framework for the development of such skills (for more information about the intervention, see [18,20–22]).

To be able to ascertain whether CT skills are being properly developed or can effectively predict performance in higher education, we need reliable, valid assessment tools. Fortunately, this assessment is available. As an example, we used a CT test— PENCRISAL [23,24]—to assess a number of first-term psychology undergraduate students, during the academic years 2011–2012 to 2015–2016, as part of the CT subject area, which forms part of a subject. PENCRISAL assesses five dimensions of critical thinking: deductive reasoning (DR), inductive reasoning (IR), practical reasoning (PR) or argumentation, decision making (DM) and problem solving (PS). We also obtained students' average academic grades and average university entrance grades. In addition, we considered some of the main dimensions of achievement motivation, using the Manassero Achievement Motivation scale [25]—although these were not used in this study beyond the procedures for monitoring course performance. With this information, we organize this article as part of our study of PENCRISAL's structural and criterion validity as a tool for assessing critical thinking in higher education students.

As we described at the beginning, the complex problems of today's world cannot be solved with a classical model of intelligence; these changes force citizens of this century to increasingly develop their ability to make solid decisions and to solve problems effectively. These competencies are precisely part of the fundamental CT competencies, widely studied in [18,19,26–28]. However, there are still a lack of studies that prove that the CT model is better than the classic model of intelligence in solving the problems of today's world [27]. This study aims to fill this gap. Therefore, one of the objectives of our work is to demonstrate that CT predicts academic performance well, something that has not yet been proven. The other objective of our study is to demonstrate the structural validity of our CT assessment test. We must be sure that our measurement instrument measures what it says it measures and that, in addition, it predicts the performance of our university students. For this, we

used a sample of more than 600 students, in order to test the construct validity. Furthermore, we used a sample of more than 200 students who underwent a CT instructional program, in order to test the predictive validity.

Thus, the purpose of our study is to test the structural and criterion validity of our CT assessment test and, therefore, to show the degree to which CT can account for academic performance. It is not our intention here to deal with conceptual developments or intervention procedures. However, we have cited our own works and those of relevant authors so that the interested reader can delve into these topics there. Let us now describe the methodology used in our study.

## 2. Method

### 2.1. Participants

The sample for this study was composed of first-year undergraduate students at the University of Salamanca studying Psychology of Thought (a CT subject area that explains what CT is). These students were assessed at the beginning and the end of this course, which focuses on developing critical thinking skills. Data were collected for 5 years, from the academic years 2011–2012 to 2015–2016, within the CT subject area.

The sample consisted of 682 students, most of whom were women (60.2%), aged 18 to 35 ($M$ = 19.02, $SD$ = 2.17). To examine the structural validity of the PENCRISAL test, a sub-sample of 242 students with academic performance information was taken to assess criterion validity. The majority of this sub-sample were also women (51.2%), with ages ranging from 18 to 28 ($M$ = 18.67, $SD$ = 1.38).

### 2.2. Instruments

Critical Thinking Test. To measure critical thinking skills, we applied the PENCRISAL test [23,24]. This consists of 35 situations relating to production problems with an open-ended response format. It has five factors: deductive reasoning, inductive reasoning, practical reasoning, decision making and problem solving; each factor has 7 items. There is currently a Peruvian adaptation [29] and a Brazilian-Portuguese adaptation [30].

The items in each of the factors reflect the most representative structures of fundamental critical thinking skills; these are briefly described below. The items that make up the deductive reasoning factor assess the most important forms of reasoning: propositional reasoning (four items) and categorical reasoning (three items). Formal reasoning is less frequent than practical and inductive reasoning, but is used to some extent. The inductive reasoning factor includes: (1) causal reasoning (three items); (2) analogical reasoning (two items); (3) hypothetical reasoning (one item); and (4) inductive generalisations (one item). The decision-making dimension assesses the use of general decision procedures, which requires making accurate probability judgements and using appropriate heuristics to make sound decisions. Two general situations are included here, whereby we need to proceed in a certain way in order to reach the best decision. In the other five situations, we need to identify the main heuristics and the biases they produce. Lastly, as with the decision-making items, the problem-solving items are divided into *general* problems (four items) and *specific* problems (three items); these will require specific solution strategies to be employed. Both the decision-making and problem-solving factors promote the performance of general decision and solution processes, with the aim of stimulating the use of the strategies necessary for planning around a problem. Meta-knowledge and the awareness of thought processes are where action is planned, directed, organised and regulated.

The format of the items is open-ended, meaning that participants must respond to a specific question by adding a justification for their answer(s). For this reason, standardised grading criteria are used to assign values ranging between 0 and 2, depending on the quality of the response. The test provides a total score for critical thinking skills and five other scores for the five factors. The range of values is 0–70 as the maximum total score for the test, with 0–14 points allocated for each of the five dimensions. Reliability measures demonstrate adequate levels of accuracy according to the scoring procedures, with the

lowest Cronbach Alpha values at 0.632 and a test–retest correlation of 0.786 [23]. The PENCRISAL test was administered as a computerised version, via the internet, using the evaluation platform: SelectSurvey.NET V5: (https://www.classapps.com/product_ssv5.aspx (accessed on 1 May 2016)). PENCRISAL is described in more detail in [24].

*Academic performance.* For the analysis of criterion validity, some of the participating students' academic records were collected—specifically, their *average academic score* and *university admission score*. The latter was obtained by adding together the average baccalaureate score (weighted at 60%) and average university entrance exam grade (40%). This gives the admission grade, which needs to be at least 5 points for students to be accepted into university, and can be up to 10 points.

### 2.3. Procedure

The study was conducted over five academic years—from 2011–2012 to 2015–2016—with first-year psychology undergraduates at the University of Salamanca. The CT instruction programme was applied during the second term as part of the Psychology of Thought course (CT subject area). To obtain a baseline of students' performance in critical thinking, we administered the PENCRISAL test one week before the start of the course (pre-treatment measurements: PENCRISAL_M1). The CT programme then ran for four months. A week after the end of the course, a second measurement was taken (post-treatment measurement: PENCRISAL_M2) of critical thinking skills, using the same test. Students were informed of the aims of the study and gave their informed consent for their data to be used for research purposes and to be analysed anonymously as part of this study. The information on students' university entrance grades and their average grade at the end of the degree was provided by the faculty's administrative services team.

### 3. Results

Table 1 shows the distribution of students' scores from the five PENCRISAL subtests by adding together the marks for their respective items. Along with the minimum and maximum values, we show the mean and standard deviation, skewness distribution coefficients and kurtosis. These values take into account the two timepoints when PENCRISAL was applied, i.e., at the beginning (pre) and at the end (post) of the course.

**Table 1.** Distribution of the students' scores in the five subtests at the two evaluation timepoints.

| Subtest | N | Min. | Max. | Mean | SD | Skewn. | Kurt. |
|---|---|---|---|---|---|---|---|
| PencriDR_pre | 682 | 0 | 10 | 3.05 | 1.85 | 0.541 | 0.174 |
| PencriIR_pre | 682 | 0 | 10 | 4.49 | 1.46 | 0.120 | 0.071 |
| PencriPS_pre | 682 | 0 | 11 | 6.46 | 2.12 | −0.355 | −0.372 |
| PencriPR_pre | 682 | 0 | 13 | 6.48 | 2.66 | 0.026 | −0.663 |
| PencriDM_pre | 682 | 1 | 14 | 6.67 | 1.97 | −0.015 | −0.204 |
| PencriDR_post | 682 | 0 | 13 | 4.55 | 2.36 | 0.417 | −0.100 |
| PencriIR_post | 682 | 0 | 11 | 5.51 | 1.76 | 0.209 | −0.034 |
| PencriPS_post | 682 | 0 | 11 | 6.34 | 2.42 | −0.341 | −0.477 |
| PencriPR_post | 682 | 0 | 14 | 8.51 | 2.48 | −0.386 | 0.054 |
| PencriDM_post | 682 | 1 | 14 | 8.22 | 2.13 | −0.260 | −0.211 |

(Pencri: pencrisal test; DR: deductive reasoning; IR: inductive reasoning; PS: problem solving; PR: practical reasoning or argumentation; DM: decision making).

As Table 1 shows, the students' scores for all five critical thinking subtests were low, with scores of zero or just one point. The post-course scores were higher than the pre-course scores, meaning that at the end of the course, some students had improved their performance in the subtests. The only exception was the problem-solving (PS) subtest, as

the average at both timepoints was very similar, and even slightly higher in the pre-course measurement (before starting the subject). In addition, the skewness and kurtosis indices for the distribution of the results in the sample fit a Gaussian distribution, as they were always less than unity. Lastly, the standard deviation of the results for the five subtests rose in the post-course measurements. This means there was more heterogeneity in the students' scores at the end of the course, which may be because some students benefitted from the critical thinking lessons and others did not. This did not occur in the PS subtest, as the variance values were similar in pre-course and post-course measurements.

To analyse the changes in students' scores in each subtest, pre- and post-course, we calculated the mean differences using the t-test for dependent samples (paired-samples t-test). To appreciate the magnitude of the difference between two moments, Cohen's d was estimated. Most of the differences are statistically significant: PencriDR (t = −15.209, df = 681, $p < 0.001$, d = −0.582); PencriIR (t = −12.067, df = 681, $p < 0.001$, d = −0.483); PencriPR (t = −18.237, df = 681, $p < 0.001$, d = −0.698); and PencriDM (t = −15.847, df = 681, $p < 0.001$, d = −0.607). In all four of these subtests, Cohen´d presents near 0.50 or a medium effect size [31]. The *p* and d values indicate clearer improvement in the scores in four PENCRISAL subtests. In the problem-solving subtest, there was a slightly higher mean in the pre-course test, but it was not statistically significant and a very small effect size was observed (t = 1.326, df = 681, $p = 0.093$, d = 0.051).

Bearing in mind one of the central objectives of our study, Table 2 shows the results of PENCRISAL's structural validity. Taking the test authors' theoretical model, the indices of fit for the unidimensional model using the AMOS programme are presented [32,33]. This is because each of the five tests assesses a different dimension or cognitive process relevant to critical thinking, all adding up to one overall score. In other words, the PENCRISAL test proposes a measure of general critical thinking ability by taking the diversity of cognitive aspects presented in the five subtests: deductive reasoning, inductive reasoning, practical reasoning, decision making and problem solving. As pre-course and post-course data were available, Table 2 shows the indices of fit for these two test timepoints (PENCRISAL_M1 and PENCRISAL_M2), indicating the confidence intervals (CI) for the RMSEA coefficient.

**Table 2.** Fit indices for the one-dimensional model of PENCRISAL.

| Indices | $\chi^2$/df | TLI | CFI | RMSEA [90% CI] |
|---|---|---|---|---|
| PENCRISAL_M1 | 3.675 | 0.946 | 0.945 | 0.06 [0.03, 0.09] |
| PENCRISAL_M2 | 1.504 | 0.982 | 0.991 | 0.03 [0.00, 0.06] |

There were good indices of fit for both applications of PENCRISAL, which were slightly higher at the end of the course (timepoint 2). In both cases, CMIN/DF was below 5.0, the TLI and CFI indices were above or very close to 0.95, and RMSEA was below 0.08 (in the pre-course measurements, the RMSEA 90% confidence interval increased to 0.09). These indices are within the required parameters [34]. One potential improvement for the PENCRISAL model at the beginning of the course is indicated by the RMSEA upper confidence interval being below 0.08 if we correlate the errors for the deductive reasoning and inductive reasoning subtests. We will return to this in the discussion.

These indices highlight the fact that the five subtests combine to give one overall score—understood here as critical thinking. The contribution of each subtest to the overall factor is shown in Figure 1, with the two timepoints separated.

The factor weightings of some subtests increased between the start and end of the course, although in two cases, they remained the same (deductive reasoning) or decreased (practical reasoning). The deductive reasoning subtest was less strongly related to the latent critical thinking variable at both assessment timepoints. Both the inductive reasoning and deductive reasoning subtests were less strongly related to the general critical thinking factor than the other three subtests (practical reasoning, decision making and problem solving).

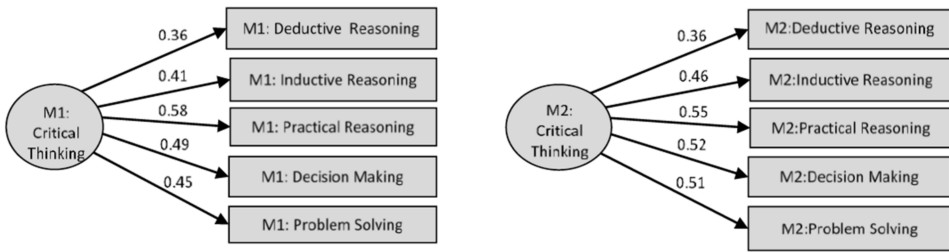

**Figure 1.** Weights of the five subtests in the general factor from PENCRISAL for the two test timepoints.

Turning to the second objective of this study—criterion validity—Table 3 shows the distribution of students' results from the pre-course and post-course PENCRISAL tests, admission scores and final average course scores. In addition to maximum and minimum values, the means, standard deviations, skewness and kurtosis, and correlations between variables are included in the table. The PENCRISAL score is the sum of the students' scores in the five subtests, and these results are from a subsample of 242 students.

**Table 3.** Distribution of the PENCRISAL results in the criterion variables and their relationships.

| Variables | Min–Max | Mean | SD | Skewn. | Kurt. | AS | CS |
|---|---|---|---|---|---|---|---|
| PENCRISAL_M1 | 12–45 | 28.65 | 5.99 | −0.14 | −0.12 | −13 * | 0.20 ** |
| PENCRISAL_M2 | 14–52 | 34.30 | 7.38 | 0.08 | 0.02 | 0.07 | 0.32 *** |
| Admission score (AS) | 5.16–9.78 | 7.55 | 0.78 | −0.21 | 0.40 | | 0.40 *** |
| Course score (CS) | 6.39–8.92 | 7.40 | 0.51 | 0.32 | −0.27 | 0.40 *** | - |

Legend: * $p < 0.05$; ** $p < 0.01$; *** $p < 0.001$.

All of the correlations were statistically significant—which is in line with our second objective. The values for the correlations between the criterion variables were the highest, which may indicate intellectual baseline differences and cognitive improvements over the years. These data require further consideration, which we cover in the following section.

## 4. Discussion and Conclusions

Overall, the results are in line with our initial proposals. Critical thinking (CT) can account reasonably well for academic performance. Additionally, the test used to measure CT shows strong unidimensional structural validity, with an overall CT factor supported by the core dimensions of CT (deductive reasoning, inductive reasoning, practical reasoning, decision making and problem solving). In addition, the statistically significant correlations between the total scores in the pre-course test and post-course test and the criterion variables support our belief that CT can be a good predictor of academic performance.

Having an assessment of the level of CT before and after studying the topic provides useful information on the potential for improving these basic CT competences. With one exception, we were able to observe an increase in post-course versus pre-course scores in most factors or subtests. This reinforces the idea that CT can be improved with training and practice. Only in regard to the problem-solving aspect (where the scores from the post-course test were no better than for the pre-course test) was this improvement not seen. Bearing in mind the sample of classes measured (2011–2016) this lack of improvement is probably due to the way this aspect operated at the time. Decision making and problem solving both employ general strategies that are hard to separate. Additionally, at the time the study took place, the activities used to improve these skills were not yet able to distinguish between them sufficiently well. We have since managed to eliminate this overlap.

There are some less robust data that need to be substantiated. A correction was made to the pre-course measurement to reduce the RMSEA index in order for it to better align

with the nature of the sample. The DR and IR subtests are more formal skills than the others and this makes them less sensitive to change because they are more difficult to apply and generalise. This results in a weaker correlation with the overall CT factor. It is important to remember that the CT test items are all problems that need to be answered by using, applying and generalising those specific processes; formal processes are less flexible and less easily modified for use outside their essentially algorithmic domain.

The indices of fit support the existence of a unidimensional model, which also worked better at the end of the course; the improvement at the end of the course may reinforce all dimensions of CT as a whole. Good performance is not possible without all CT core competencies working together. This may explain the unidimensional nature of the PENCRISAL test in terms of its structural validity.

In terms of criterion validity, we found significant relationships between CT and academic performance, and an even stronger relationship between this criterion variable and the university admissions measurement. As we noted above, these data go in the direction that we expected, based on our initial approach. However, there was no relationship between CT and university entry requirements. One way to interpret this lack of relationship may be the fact that the level of CT measured at the start of the course was fairly low in relation to the test's reference standards. Moreover, whilst post-course measurements of CT demonstrated an increase in the level of CT, they were still too limited to capture the relationship that should exist with the entry requirements. CT skills are complex and require significant levels of expertise in order to be able to capture correlations with measures of a different intellectual nature—such as those that may underlie a university entrance score, which is the result of several years of schooling. In a study into how permanent the change in CT is following teaching and measured four years later, we saw greater improvement than immediately after the intervention. This improvement is attributable not only to the CT programme, but also to the experience and education gained throughout those four years at university [20], although this is not always the case (see [35]). We can therefore say that in this study, the relationships between CT and academic performance are easier to see (as they are the result of experience and education) than those between CT and the cut-off or selection measures for university admission, which capture a more stable and experience-independent threshold.

A number of implications emerge from our study. First, in order to be able to use CT levels as a predictor, we need levels that are above the average of the test scales used. Second, it is possible to improve CT competencies using the right instructional tools and given sufficient practice and procedural work. Lastly, we need to take CT measurements in order to ascertain our starting point, and thus, see what has been achieved. If we believe that today's world requires these complex skills, we also need to ascertain the degree to which they are available from the outset. Without such an assessment, we cannot see how far we have come.

**Author Contributions:** Conceptualization, S.F.R., C.S. and L.S.A.; methodology, S.F.R. and L.S.A.; software, S.F.R. and C.S.; validation, S.F.R., C.S. and L.S.A.; formal analysis, S.F.R., C.S. and L.S.A.; investigation, S.F.R., C.S. and L.S.A.; resources, S.F.R., C.S. and L.S.A.; data curation, S.F.R. and L.S.A.; writing—original draft preparation, S.F.R., C.S. and L.S.A.; writing—review and editing, S.F.R., C.S. and L.S.A.; visualization, S.F.R. and C.S.; supervision, S.F.R. and C.S.; project administration, S.F.R. and C.S.; funding acquisition, S.F.R. and C.S. All authors have read and agreed to the published version of the manuscript.

**Funding:** This research received no external funding.

**Institutional Review Board Statement:** Not applicable.

**Informed Consent Statement:** Informed consent was obtained from all subjects involved in the study.

**Data Availability Statement:** Not applicable.

**Conflicts of Interest:** The authors declare no conflict of interest.

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
