# Peer review of "The Role of Critical Thinking in Predicting and Improving Academic Performance"

_sustainability, doi:10.3390/su15021527_

Round 1

Reviewer 1 Report

This paper has potential but as it stands is not very enlighening.

The main reason is that we are measuring CT which is never well defined--we have a number of elments perhaps but we need an overall concept--it is not there. 

I still think that the topic is not well-defined

It remains unclear to me what precisely companies find lacking (p.2 line 52). Undoubtedly, this would vary greatly but no recognition that is made. This might be where the topic could be made clear even with good examples, then we might have some idea as to how to address the matter and on what level (p.6 line 253-258).

Is it for instance an incapacity to make decisions or is it a weakness in analyzing issues or….

At the moment the concept CT is far too generic to get any fruitful result.

This characterizes the paper more widely: it has a good deal of somewhat interesting observations but they don’t really find a focus in this paper.

Similarly, use of PENCRISAL seems to be a bit like a crystal ball without any context of how and where this instrument was used and to what effect.

So, to make this article worthwhile and publishable it needs to be great trimmed and focused before any university would find it useful.

I wish I could be more positive but the paper, though promising, does not deliver.

Author Response

The new version of the manuscript includes new paragraphs in order to respond to most of the reviewers' suggestions. However, we will try to justify the comments indicated by reviewer 1.

The objectives of our work are to predict academic performance and demonstrate the construct validity of our test. The manuscript is not intended for conceptual development and discussion of critical thinking, nor critical thinking (CT) instruction. We have published papers for these purposes and so have relevant authors in the field. In addition, in the works cited in the manuscript, the conception and conceptual approach on which we are based can be clearly seen. However, as we have said, we have added some paragraphs to illuminate the conceptual basis and incorporated new references that shed more light on this reviewer's comment.

The critical thinking test that we have psychometrically validated would be very difficult for us to have obtained good reliability and validity indices, if it were not conceptually reasonably defined. In the validation publications we broadly define what we mean by TC. Again, the conceptual treatment, already published elsewhere, is not the object of study in this work. We must remember, however, that the test developed consists of five dimensions, which contemplate and measure the fundamental competences of TC, assumed and considered by the relevant authors of the field.

In research on education, for a long time, the disconnection that exists between the training we give our students and the training that companies demand has been verified. What they need are more horizontal than vertical competencies, just the TC competencies. In one of the papers used as the basis for our manuscript (Saiz, Rivas & Almeida, 2020) this problem is exposed, which reviewer 1 comments on (p.6, line 52). The dimensions or subtest of the TC test are described and, in some cases, with examples, in the validation and conceptual foundation publications of the same. This responds to the other request from reviewer 1 (p.6, line 253-258)

Everything related to the nature of the fundamental TC competencies, such as the ability to make decisions, is extensively developed in other publications that we have carried out, and which are cited in the new version of the manuscript. Once again, in this paper it is not our aim to carry out a conceptual treatment like the one requested by reviewer 1, although we have incorporated some paragraphs to help in this direction.

Regarding the results, the reviewer tells us that the concept of TC is generic and that, therefore, the results fail to be fruitful. The TC measurement is carried out with a psychometrically validated test and its dimensions are specifically defined, in addition to its conceptual foundation, but all this is found in other already published works. In the present study we only summarize the dimensions in order that the reader can understand the results. If a reader needs to consult some detail in greater depth, he has the validation publications of the test.

The focus or objective of our work is clear, answering the problem of whether the CT can be used to predict academic performance, once developed in our students. This prognostic capacity of TC is unknown, and we have managed to show that it does have it. We have identified an important problem (so we believe) in the field and we have given an answer to it. This is one of the focuses of the work, although not the only one, as we have already said.

Reviewer 1 tells us that the use of the pencrisal is something of a crystal ball without any context for how and where it was used, and to what effect. A psychometrically validated test needs a precise conceptual definition, it is also applied in the context of higher education, and finally reasonably positive results are obtained as stated, we wonder if this is "a crystal ball", we will answer what a welcome that crystal ball is.

To reviewer 1's assertion that the article must be well defined and focused to be useful, we can only respond to the following. In the manuscript we propose some objectives that are the answer to some relevant problems in the field of TC, such as having measures with a solid construct validity, and with a predictive validity according to current times. We have carried out an evaluation of CT in a sample of more than 600 university participants, and we have instructed more than 200 university participants in CT. The results obtained support our initial proposal reasonably well. If this isn't focused and relevant work, it's probably because we don't quite understand what reviewer 1 is trying to tell us.

Finally, we thank reviewer 1 for his comments as they always improve our work.

Reviewer 2 Report

Review Report

Article title:

The Role of Critical Thinking in Predicting and Improving Academic Performance

This manuscript reports on the result of the study that explores the extent to which critical thinking (CT) can be used to improve academic performance. The article raises an important and well-focused research topic. It is publishable subject to revision. The problems of this manuscript are detailed below.

Specific comments:

1.   The introduction should include the objective of the study.

2.  The research questions and specific hypotheses being tested need to be clearly mentioned at the end of the “Introduction” section.

3.   The current state of the object of research should be reviewed carefully, and key publications should be cited. A literature review regarding what is known in the field of critical thinking and its use in predicting and improving academic performance is much needed and required.

4.   It would be useful to inform the reader a bit more about the content of teaching. Present, in more detail, the exact procedures that were followed during teaching.

5. The section “Discussion and conclusions” needs to be better unfolded, highlighting is new in the research and how the outcomes of the study can be used in practice. The authors should discuss the results and how they can be interpreted in terms of the research hypotheses. Have the results been compared with literature? Have you found any similarities or discrepancies with previously published data? What are the benefits of the study for the future?

6.  References must be numbered in order of their appearance in the text and listed individually at the end of the manuscript. In the text, the reference numbers should be placed in square brackets [ ] and positioned before the punctuation.

Kind regards,

The Reviewer

Author Response

La nueva versión del manuscrito incluye nuevos párrafos para responder a la mayoría de las sugerencias de los revisores. No obstante, intentaremos justificar los comentarios señalados por el revisor 2.

-El revisor 2 nos pide que incluyamos los objetivos del estudio en la introducción. En el nuevo manuscrito, y en los nuevos párrafos, los objetivos se incluyen de forma más específica.

-Al final de la introducción se especifica y resume el problema de investigación y la propuesta de solución según lo indicado por el revisor 2.

-Reviewer 2 tells us that a careful review of the key publications should be done and cited. Regarding the objectives of our work, we can say that the relevant works have been cited, which are very few, since the problem of whether critical thinking (CT) can predict academic performance after an instruction has not been dealt with, with the exception of the empirical work done by us. We would have liked to review other works, but we have not found them, at least in the same context in which they are treated here.

-In the new version of the manuscript, some paragraphs and references are added, in order to help the reader in understanding our conceptual approach, and regarding the instruction in CT.

-As we have said, there are no studies with which we can compare our work, because we are addressing a new problem that has not been investigated empirically until now. We would have liked to have other studies, because this always enriches the investigation, but we have not found them. On the other hand, the hypotheses of our work have been discussed in the discussion section. We postulate that the CT predicts academic achievement and we have obtained reasonable support for our prediction, as well as the general factorial structure of the test. We have also explained the data that does not support our predictions, specifically, in some of the TC dimensions. The sample used was instructed with an earlier version of our instructional program, but not with the one that was recently improved. For this reason, those dimensions did not work as well with the previous version of the program. In the discussion and conclusions section there are other considerations that we believe include everything suggested by reviewer 2.

-Las referencias se han ordenado alfabéticamente, siguiendo las instrucciones de la revista. No obstante, si hay algún error formal en este apartado, lo corregiremos.

Finalmente, agradecemos al revisor 2 por sus comentarios, ya que siempre mejoran nuestro trabajo.

Round 2

Reviewer 2 Report

Review Report 2

The authors addressed most of my concerns partially.

The current state of the object of research should be reviewed carefully, and key publications should be cited. A literature review regarding what is known in the field of critical thinking and its use in predicting and improving academic performance is much needed and required.

For example:

Jenkins E.K.: The Significant Role of Critical Thinking in Predicting Auditing Students' Performance, https://doi.org/10.1080/08832329809601644

Fernando A.D’Alessio; Beatrice E. Avolio; Vincent Charles Studying the impact of critical thinking on the academic performance of executive MBA students, https://doi.org/10.1016/j.tsc.2019.02.002

Xuezhu Ren; YanTong; PengPeng; TengfeiWang Critical thinking predicts academic performance beyond general cognitive ability: Evidence from adults and children, https://doi.org/10.1016/j.intell.2020.101487

etc.

Text on Figure 1 is not readable.

References must be numbered in order of their appearance in the text and listed individually at the end of the manuscript. In the text, the reference numbers should be placed in square brackets [ ] and positioned before the punctuation. Instructions for Authors available on: https://www.mdpi.com/journal/sustainability/instructions.

Author Response

Once again, we thank reviewer 2 for his comments, which have contributed to improving our manuscript once again.

Reviewer 2 asks us to review the most representative works on critical thinking (CT) and academic performance. Certainly, there are publications in this regard, which have already been analyzed and studied by us in other publications for other research purposes. In the present work our objectives are other. As we stated in lines 155-157, there is a lack of studies that prove the role of CT in everyday and academic performance. It is possible that our proposal is not clear enough, for this reason reviewer 2 asks us to cite and compare the studies that exist on CT and academic performance. Certainly, studies of this kind exist, but they are general and correlational or regression in nature. They are studies that measure CT, intelligence and academic performance (and other variables) and study the relationship that exists or the predictive capacity of one over the other, through regression weights. However, our work is different in several aspects, which allows us to affirm that there are few relevant studies of this type. Why is our work different? Because it has four unique features that we present below:

-The first singular characteristic of our study is the instruction. We do not only measure CT, the study participants go through our training program, and we carry out a CT evaluation before and after it (it should be remembered that this intervention lasts between 50-55 hours of face-to-face work in Classroom). In this way, we can know the level of CT after the instruction, we can identify the change produced, that is, we manipulate the CT to be the best. In this way, we can predict academic performance from the different actual levels of CT. We do not measure the baseline CT alone, we also evaluate the TC after its improvement. And we are unaware of studies of this kind with these objectives.

-The second singular characteristic of our study consists in evaluating the validity of the specific construct for our sample, which consists of more than 600 participants, and what do we obtain in our confirmatory factor analysis? Well, a general CT factor, with subfactors, but the important thing is that we found one, CT, which explains most of the variance. We are also unaware of studies that perform this factorial demonstration of the CT measure to ensure that they have that general factor that is supposed to predict academic performance. What we find in these studies is the application of some of the CT tests that exist without further ado.

-The third singular characteristic of our study consists of the measurement of CT. We have developed a test consisting of items that are complex, personal and professional problems that must be answered with open answers. In this way, and through the task analysis technique, which is exposed in the publication on the theoretical foundation of the test, we make sure that each answer to each problem is of a type. If the problem requires, for example, the use of decision or problem-solving strategies, we will know which strategy is being used in each case and not others. The standardized CT tests used in most studies are the Facione test, the Watson and Glaser test, or the Cornell de Ennis test. These tests cannot differentiate processes, because they are really response recognition tests, which prevent us from knowing which thought process is responsible for that response. These tests also consist of items that are too academic and artificial. If a CT test does not pose complex problems or complex tasks to be solved, the most substantial and complex aspects of CT are not being captured. There are only two tests, that we know of, that meet these current CT measurement requirements, the Halpern HCTA test (Halpern, 2006-2010-2018; Halpern & Dunn, 2021) and ours, the PENCRISAL. We know the Halpern test very well because we try to validate it in our language:

Nieto, A.M., Saiz, C. & Orgaz, B. (2009). Análisis de la propiedades psicométricas de la versión española del HCTAES-Test de Halpern para la evaluación del pensamiento crítico mediante situaciones cotidianas. Revista Electrónica de Metodología Aplicada, 14 (1), 1-15.

Nieto, A.M. & Saiz, C. (2008). Evaluation of Halpern’s “Structural Component” for Improving Critical Thinking. The Spanish Journal of Psychology, 11 (1), 266-274.

As a consequence of these studies, we adopted one of the innovative characteristics of the Diane Halpern test, for the development of our test, that is, using complex, daily and professional problems as items. Asking that these kinds of problems be solved substantially changes the evaluation of the CT in the sense of guaranteeing a much more powerful validity, because it assures us, compared to other tests, that we are really measuring what we want to measure. Our test incorporates task analysis, which allows us to ensure that open-ended responses reveal the use of a specific thought process. This technique is not found in any other CT test. Obviously, we are not aware of studies that use at least the HCTA test for the same purposes as those that have guided us in this work. Without a test, so to speak, of a new generation, it is not possible to know what goes through the minds of the participants when they answer multiple choice questions. A lot of effort is needed to improve CT and we need to measure it with sufficient guarantees that we measure what we want to measure. We cannot afford to spend so much time on training and then use measures that do not give us the information we need.

-The fourth and final unique feature of our study is determined by our approach to CT. If we consider that CT is to solve problems effectively, through the best explanation, in order to produce a change in reality, then we must use instructional procedures in CT, based on performance and behavior. And if we want to know if these changes have occurred, we must use behavioral evaluation methods, consisting of carrying out tasks, in execution, not in self-reports, not in multiple-choice tests, not in the calibrations made by the participants of a study on its performance. For this reason, we have wanted, in the present study, to make sure that the CT variable is the one that explains most of the variance, as confirmed by the confirmatory factor analysis. Furthermore, once we manipulate CT through instruction, we can see, before and after instruction, how CT predicts academic performance, as our results confirm for the most part. In short, the essence of our conception rests on the process of explanation or causality, that is, we are not interested in identifying correlations, applying tests that measure different variables and seeing how they are related. We rather look for the determinants or the causes that explain or solve a problem. Due to all of the above, we are really unaware of the studies that we can compare to our approach and objectives in the present work.

These are the reasons why we have not compared our study and our data with other generic studies that exist.

Reviewer 2 informs us that in figure 1 the text is not read. We have edited that figure to solve this problem.

Reviewer 2 tells us that we do not follow the citation rules of the journal. We have corrected the citations and references to conform to these standards.

We have seen that the references for the validation and foundation of our test were not in the references. These two references have been included in the corrected version of the manuscript.

Once again, many thanks to reviewer 2 for his help in improving the present manuscript.